# Arousal state transitions occlude sensory-evoked neurovascular coupling in neonatal mice

Kyle W. Gheres [1], Hayreddin S. Ünsal [2,3], Xu Han [1], Qingguang Zhang [4], Kevin L. Turner [2], Nanyin Zhang [1,2,5,6] & Patrick J. Drew [1,2,4,5,6,7 ✉]

In the adult sensory cortex, increases in neural activity elicited by sensory stimulation usually drive vasodilation mediated by neurovascular coupling. However, whether neurovascular coupling is the same in neonatal animals as adults is controversial, as both canonical and inverted responses have been observed. We investigated the nature of neurovascular coupling in unanesthetized neonatal mice using optical imaging, electrophysiology, and BOLD fMRI. We find in neonatal (postnatal day 15, P15) mice, sensory stimulation induces a small increase in blood volume/BOLD signal, often followed by a large decrease in blood volume. An examination of arousal state of the mice revealed that neonatal mice were asleep a substantial fraction of the time, and that stimulation caused the animal to awaken. As cortical blood volume is much higher during REM and NREM sleep than the awake state, awakening occludes any sensory-evoked neurovascular coupling. When neonatal mice are stimulated during an awake period, they showed relatively normal (but slowed) neurovascular coupling, showing that that the typically observed constriction is due to arousal state changes. These result show that sleep-related vascular changes dominate over any sensory-evoked changes, and hemodynamic measures need to be considered in the context of arousal state changes.

[1] Molecular Cellular and Integrative Bioscience program, The Pennsylvania State University, University Park, PA 16802, USA. [2] Department of Biomedical Engineering, The Pennsylvania State University, University Park, PA 16802, USA. [3] Department of Electrical and Electronics Engineering, Abdullah Gul University, Kayseri, Türkiye. [4] Department of Engineering Science and Mechanics, The Pennsylvania State University, University Park, PA 16802, USA. [5] Center for Neural Engineering, The Pennsylvania State University, University Park, PA 16802, USA. [6] Center for Neurotechnology in Mental Health Research, The Pennsylvania State University, University Park, PA 16802, USA. [7] Departments of Neurosurgery and Biology, The Pennsylvania State University, University Park, PA 16802, USA. ✉email: pjd17@psu.edu

The first postnatal month of rodent life is a period of rapid brain changes, particularly in the cortex where sensory circuits undergo maturation and refinement of their neural responses[1,2]. In addition to the neural changes occurring at this time, there is a refinement of the cerebral vasculature anatomy[3–8]. In neonatal and infant rodents (before postnatal day 20, P20) and humans, several studies have found that sensory stimulation drives a large decrease in blood volume and oxygenation[9–12] several seconds after stimulation. This finding is in contrast to adult animals and humans, in which increases in neural activity drive vasodilation of the nearby vasculature (known as neurovascular coupling) via a variety of mechanisms[13,14]. However, other studies have found a positive, but slower hemodynamic responses in neonatal rodents and infants[15,16]. As the GABA reversal potential becomes inhibitory in visual cortical neurons by P3[17] and in the somatosensory cortex by at least P7[18], this altered neurovascular coupling cannot be attributed to the excitatory effects of interneurons. Resolving the nature of neurovascular coupling in the neonatal brain is important, not only for the interpretation of functional imaging signals in neonates, but also because the pattern in blood flow is an important signal in the assembly of the vasculature[3,5,19,20].

While hemodynamic signals can not only be driven by sensory-evoked increase in neural activity and/or volitional movements[21–24], they are also strongly affected by arousal-state changes. In both humans and primates, arousal-state changes have large effects on hemodynamic signals across the cortex[25–27]. In rodents, there is a profound increase in blood volume during nonrapid eye movement (NREM) and rapid eye movement (REM) sleep as compared to the awake condition[28,29]. The blood volume increases during sleep are several times larger than the increases evoked in awake animals, and awaking causes profound decreases in blood volume within a few seconds of the arousal transition[29,30]. These sleep–wake-related blood volume changes are relevant for hemodynamic imaging in neonates because neonatal rodents spend a large fraction of their time asleep[31,32], and neonatal sleep is substantially more fragmented than in the adult[33].

Here we ask how neurovascular coupling changes over development using optical imaging, BOLD fMRI, and electrophysiology in unanesthetized mice. We found that during the first postnatal weeks, stimulus-evoked hemodynamic responses emerge approximately at P14 and increase in amplitude in the following postnatal period, but are usually accompanied by a large poststimulus constriction. We find that head-fixed mice rapidly transition between sleep and wake, and that these arousal-state changes drive large, low-frequency changes in blood volume. Segregating responses by pre-stimulus arousal state in P15 mice showed that neurovascular coupling is positive in the awake neonatal mouse and that stimulation can cause awakenings whose hemodynamics changes can obscure any sensory-evoked responses. Thus the "inverted" hemodynamic response is not due to a difference in neurovascular coupling, but due to changes in arousal states.

## Results

We investigated the hemodynamic response to somatosensory stimulus in unanesthetized mice of both sexes. Unless indicated, reported numbers are mean ± standard deviation, and for statistical analysis we used generalized linear models, which can account for within-animal correlations and multiple comparisons[34,35]. All electrophysiological and imaging experiments were acute, BOLD fMRI imaging was longitudinal (see "Methods").

**Whisker stimulus-evoked hemodynamic responses emerge in the third postnatal week.** Using wide-field optical intrinsic signal (IOS) imaging, we measured blood volume changes ($\Delta[HbT]$)

in primary somatosensory cortex of head-fixed mice ages P10 up to adulthood. Mice were head-fixed on a spherical treadmill, allowing us to detect any locomotory activity[36,37] (Fig. 1a). We measured hemodynamics signals using wide-field IOS imaging with 530 nm light through a thinned-skull window. We stimulated the left and right whiskers with brief puffs of air, and a puffer not aimed at the mouse provided an auditory stimulus, but only the contralateral whisker stimulation was analyzed here. Dilations of arteries, capillaries, and veins[38] will drive a decrease in reflected light due to the increased absorbance caused by elevated hemoglobin. Reflectance changes were converted into changes in total hemoglobin[39].

We saw spontaneous changes in blood volume across all ages (Fig. 1b–d). When we quantified the evoked hemodynamic responses to the contralateral whisker stimulation, we saw that sensory-evoked responses were nearly absent in P10 mice (Fig. 1e). Following a whisker stimulus, blood volume increases were observed in adult (~4-month-old) mice, reaching their maximum (peak response: $16.8 \pm 5.02 \mu M$) shortly following stimulation (time to peak: $1.23 \pm 0.19$ s) (Fig. 1f, g), before decaying back to baseline (full-width at half max (FWHM) of the response: $1.72 \pm 0.50$ s) (Fig. 1h), closely matching what has been seen previously in awake adult mice[21,29]. However, younger mice displayed absent or attenuated hemodynamic responses to whisker stimulus. At P10, whisker stimulation results in a small amplitude, temporally delayed hemodynamic response (Fig. 1f) which was significantly smaller than in the adult (peak amplitude: $1.70 \pm 1.087 \mu M$, LME comparison with adult: $P < 3.22 \times 10^{-11}$; average integrated response: $0.323 \pm 1.154 \mu M$, LME: $P < 1.66 \times 10^{-7}$). Hemodynamic responses to sensory stimulation at this age were also more delayed than in the adult, (time to peak: P10: $2.69 \pm 1.31$ s, LME compared to 4-month-old: $P < 0.001$; FWHM: $3.98 \pm 3.97$ s, LME compared to 4-month-old: $P < 0.062$) (Fig. 1e–g), with this delay making it difficult to dissociate stimulus-evoked changes in blood volume from any behavior associated changes following the initial stimulus. Following eye opening (postnatal day 14), contralateral whisker stimulation evoked hemodynamic changes that had more adult-like timing (Fig. 1e) (time to peak, P15: $1.14 \pm 0.35$ s, $P < 0.826$ vs. adult; P20: $1.43 \pm 0.46$ s, $P < 0.649$; P28: $1.34 \pm 0.21$ s, $P < 0.800$ and FWHM: P15: $1.83 \pm 1.27$ s, $P < 0.926$; P20: $1.72 \pm 0.60$ s, $P < 0.996$; P28: $1.92 \pm 0.98$ s, $P < 0.869$) (Fig. 1a–e) though the magnitude of the evoked hemodynamic responses was significantly smaller than those in adult animals (peak response: P15: $2.16 \pm 1.92 \mu M$; $P < 5.19 \times 10^{-11}$, P20: $5.17 \pm 2.58 \mu M$; $P < 2.54 \times 10^{-8}$, P28: $4.95 \pm 2.73 \mu M$, $P < 4.34 \times 10^{-8}$; integrated response: P15: $0.62 \pm 1.83 \mu M$, $P < 4.88 \times 10^{-7}$, P20: $2.72 \pm 2.407 \mu M$, $P < 1.26 \times 10^{-4}$, P28: $2.357 \pm 2.32 \mu M$, $P < 1.03 \times 10^{-4}$) (Fig. 1i). By 6 weeks of age, animals display adult-like hemodynamic responses following whisker stimulation (peak response: $14.98 \pm 3.58 \mu M$, $P < 0.276$, time to peak: $1.26 \pm 0.08$ s, FWHM: $2.11 \pm 0.79$ s, $P < 0.736$, avg integrated response: $8.39 \pm 2.97 \mu M$, $P < 0.824$) (Fig. 1i). Consistent with previous reports of sensory-evoked responses in neonatal mice and rats[9,11], we find that the sensory stimulation drives a small increase in blood volume, followed by a large sustained undershoot (Fig. 1j).

**Whisker stimulation evokes increases in neural firing rate.** The lack of a robust blood volume response could be due to a lack of neural responsiveness. To test this possibility, we then examined the sensory-evoked changes in neural firing in postnatal mice. We recorded multiunit spiking activity and gamma-band local field potential (LFP) power (which are most strongly correlated with the hemodynamic response[21,40,41]) in the barrel cortex of unanesthetized P10, P15, and P28 mice using 16-site linear electrode

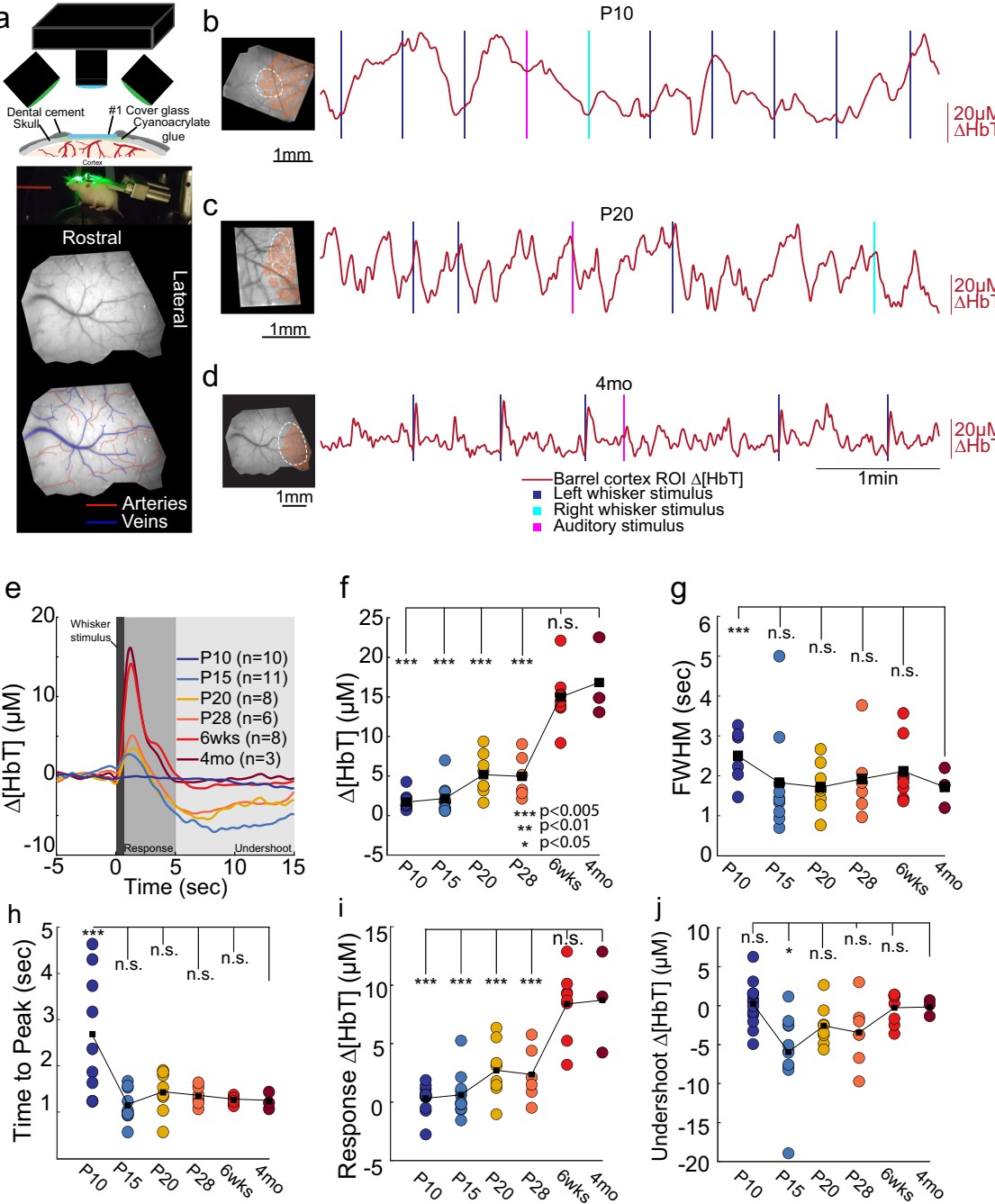

**Fig. 1 Imaging blood volume changes in the somatosensory cortex of mice at different ages. a** Schematic showing imaging setup (top), photo of mouse on ball (middle) and example window with arteries and veins outlined (bottom). **b**–**d** Example Δ[HbT] responses for three different ages of mice. Left, images of windows. The ROI is shown by dotted oval chosen the minimize overlap with the large surface veins, cytochrome-oxidase reconstructed barrel locations shown in pink. Left, time courses of Δ[HbT] during sensory stimulation. **e** Population average time course of whisker evoked blood volume changes at each age. The hemodynamic responses at early ages remain similar until ~6 weeks after birth, with an initial dilation response followed by a prolonged undershoot. **f** Peak increases in Δ[HbT] following whisker stimulus. **g** Full-width at half-maximal amplitude for each animal by age. Note that for the P10 age group, four animals lacked a clear peak and recovery in Δ[HbT] and were excluded from the FWHM and peak calculations. **h** Time following whisker stimulus onset at which peak values in (**f**) were reached. Note highly variable temporal response at P10, and adult-like timing in older animals. **i** Average increase in Δ[HbT] from 0.5 s to 3 s after stimulation onset. At younger ages, rapid reversal of dilation results in near zero or inverted hemodynamic responses for many animals. **j** Average decrease in Δ[HbT] following stimulus from 5s-15s post stimulus. Roughly half of animals P15-P28 show large, sustained decreases in blood volume beyond what is observed in adult animals. Only P10 animals used to calculate FWHM were included in undershoot average. *$P < 0.05$; **$P < 0.01$; ***$P < 0.005$.

arrays[42] (Supplementary Fig. 1). The baseline multiunit firing rates increased with age (P10: $41.3 \pm 2.3$ Hz, P15: $58.6 \pm 9.1$ Hz, P28: $75.2 \pm 10.9$ Hz, averaged across all layers), consistent with previous work[18,43]. Immediately following the stimulation onset,

we observed robust neural responses to whisker stimulation at all ages (0–100 ms following stimulation: P10: $81.2 \pm 14.0$ Hz, P15: $117.3 \pm 19.6$ Hz, P28: $140.6 \pm 15.8$ Hz). Stimulation also drove large increases in the power in the gamma band (40–100 Hz) of

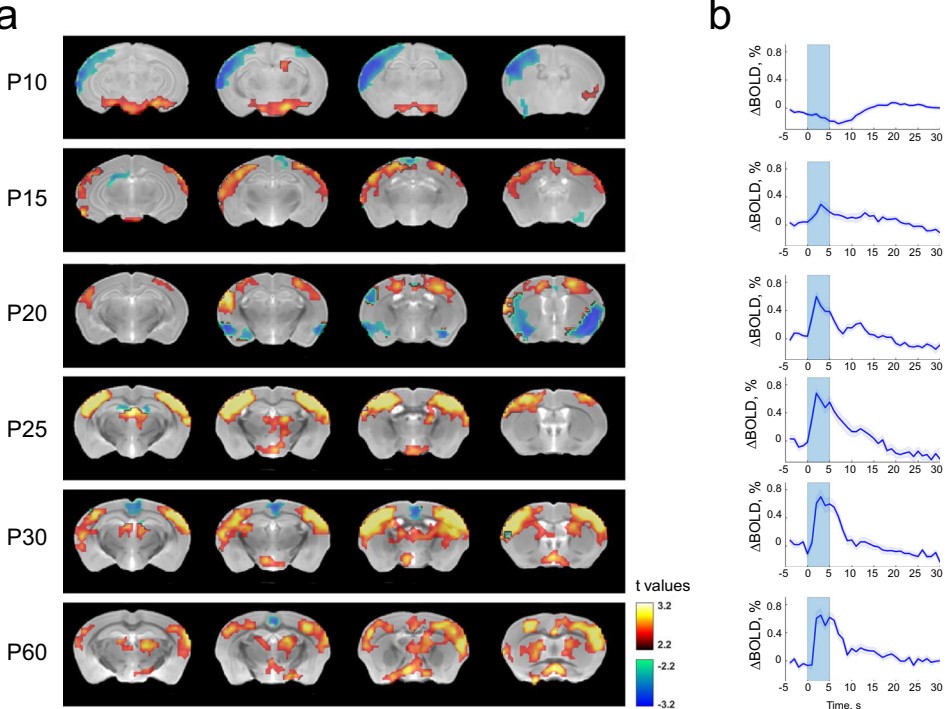

**Fig. 2 BOLD responses to whisker stimulation during postnatal development. a** BOLD activity in response to whisker stimulation at different postnatal stages. For simplicity, only brain slices with the somatosensory cortex are shown. Pseudo-colored voxels demonstrate positive (red) or negative (blue) BOLD response that is significantly different from baseline (two-sample $t$ tests, linear mixed model, $P < 0.05$, FDR corrected, $N = 18, 11, 19, 19, 22, 15$ for age P10, P15, P20, P25, P30, P60, respectively). Mouse structural images of different ages were co-registered to the Allen brain atlas with the ANTs (Advanced Normalization Tools) software. The BOLD response in the somatosensory cortex is slightly negative at P10, turns positive on P15 and gradually becomes adult-like from P20 to P60. **b** Averaged time courses of BOLD signals across activated voxels in the somatosensory cortex. Blue shaded bars indicate the on time of whisker stimulation (duration = 5 s).

(P10: $464.0 \pm 149.3\%$, P15: $412.0 \pm 339.1\%$, P28: $75.2 \pm 10.9\%$). We found that during the "steady-state" stage of whisker stimulus (200–500 ms), spiking remained elevated compared to baseline values (P10: $87.0 \pm 23.7$ Hz, P15: $138.4 \pm 37.7$ Hz, P28: $172.7 \pm 40.0$ Hz) and that gamma-band power remained elevated above baseline values (P10: $192 \pm 89\%$, P15: $92\% \pm 73\%$, P28: $818 \pm 252\%$). These results show that whisker stimulation drove robust neural activity even in the youngest of animals, and that neural activity remained increased above baseline values across the duration of the stimulus in the absence of adult-like stimulus-evoked hemodynamic changes. Our results are consistent with the literature on the responsiveness of cortical neurons to external stimuli during the neonatal period[31,43–47], and show that the lack of observed sensory-evoked vasodilation is not due to a lack of underlying neural responsiveness.

**Sensory stimulus-evoked BOLD changes are reflective of underlying hemodynamics.** While optical measures of cortical blood volume can be performed in head-fixed rodents, neuroimaging studies in young humans rely on oxygen-dependent signals measured with fMRI. We compared our whisker stimulus-evoked blood volume changes with whisker stimulus and visual stimulus-evoked BOLD changes in primary sensory cortex spanning the juvenile period in mice. Prior to eye opening, whisker stimulation evoked a slight decrease in the BOLD response (peak amplitude $= -0.17 \pm 0.03\%$), consistent with our results of increased spiking activity (i.e., likely increased oxygen consumption) and lack of CBV change. Similar to recorded hemodynamic signals, following eye opening BOLD signal amplitudes increased with age in primary somatosensory cortex (Fig. 2a, b). Relative to the BOLD amplitude in adult mice (P60, peak amplitude $= 0.56 \pm 0.07\%$), P15

mice showed a significantly smaller BOLD response (peak amplitude $= 0.19 \pm 0.06\%$; $P = 0.026$). From P20, whisker stimulation-induced BOLD response was comparable to that in adult mice (peak amplitude: P20, $0.34 \pm 0.05\%$, $P = 0.33$; P25, $0.48 \pm 0.08\%$, $P = 0.94$; P30, $0.57 \pm 0.09\%$, $P = 0.83$).

Similar results were found in the visually evoked BOLD responses in the visual pathway, including the superior colliculus (SC) and primary visual cortex (V1, Supplementary Figs. 2 and 3). P10 mice (prior to eye opening) displayed slightly negative BOLD response to the visual stimulation in both the SC and V1 (SC: peak amplitude $= -0.020 \pm 0.032\%$, V1: peak amplitude $= -0.076 \pm 0.022\%$), and the amplitudes were significantly lower than adult mice (SC: peak amplitude $= 0.225 \pm 0.028\%$, $P = 4.6 \times 10^{-8}$; V1: peak amplitude $= 0.163 \pm 0.024\%$, $P = 1.2 \times 10^{-11}$). On P14 (day of eye opening), the BOLD response became positive, but the amplitude was still smaller than adult mice (SC: peak amplitude $= 0.11 \pm 0.020\%$, $P = 3.5 \times 10^{-4}$; V1: peak amplitude $= 0.053 \pm 0.019\%$, $P = 3.0 \times 10^{-4}$). From P20, visual stimulation-induced BOLD response was not significantly different from adult mice (SC, peak amplitude: P20, $0.32 \pm 0.05\%$, $P = 0.11$; P25, $0.16 \pm 0.03\%$, $P = 0.10$; P30, $0.26 \pm 0.05\%$, $P = 0.54$; V1, peak amplitude: P20, $0.22 \pm 0.04\%$, $P = 0.26$; P25, $0.13 \pm 0.02\%$, $P = 0.31$; P30, $0.16 \pm 0.05\%$, $P = 0.98$). These results show that the dynamics and developmental time course of the BOLD responses match the blood volume responses from optical imaging experiments, and that the negative response to sensory stimulation is not solely restricted to S1.

**Younger mice show larger hemodynamic fluctuations at lower frequencies.** We then quantified how spontaneous fluctuations in blood volume in the absence of overt stimulation change over development. In very young animals, we saw large, low-frequency

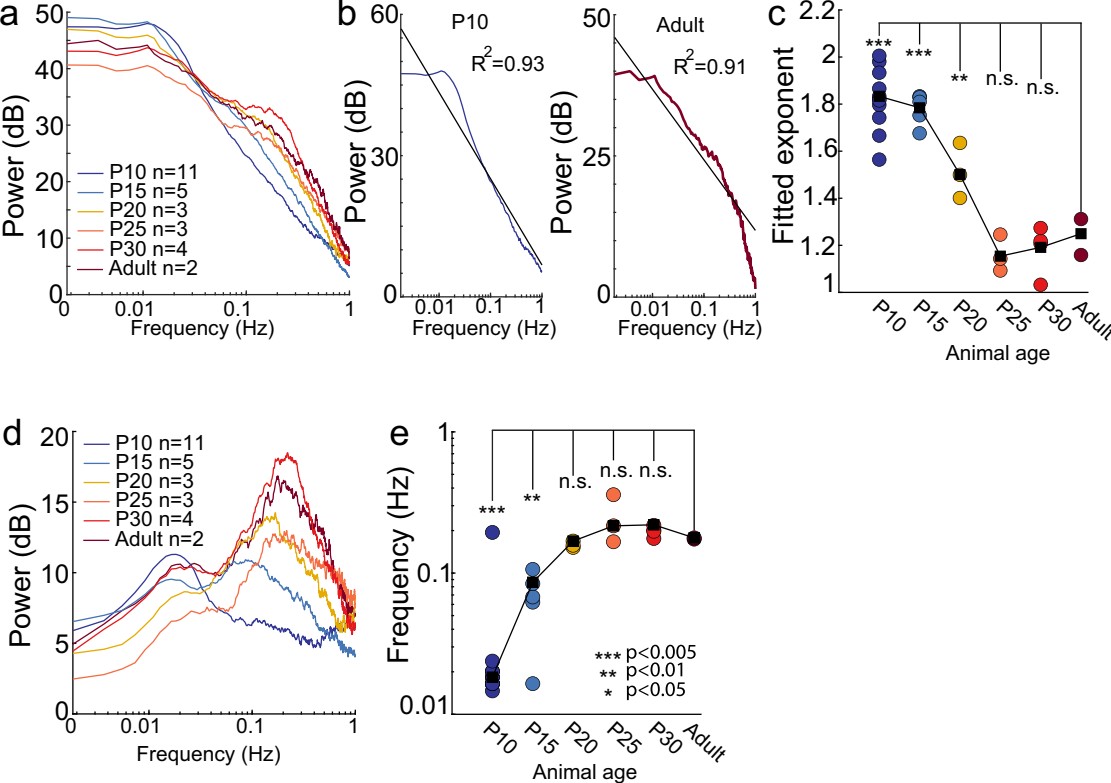

**Fig. 3 Spontaneous, ultra-slow changes in blood volume are observed at rest in juvenile mice. a** Average resting-state power spectrum of Δ[HbT] for each age. Notice a decrease in power below ~0.02 Hz and development of "vasomotion" at 0.1–0.3 Hz as mice age. **b** Examples of individual animal power spectra fit with a power law of the form $1/f^\beta$. **c** Plot of the exponents fitted to individual animal. **d** Pre-whitened power spectrum of Δ[HbT] for each age. Note the shift of the peak to higher frequencies with age. **e** Frequency of peak power of pre-whitened spontaneous Δ[HbT] power spectrum. Note the P10 peak is substantially lower than that of older mice. *$P < 0.05$; **$P < 0.01$; ***$P < 0.005$ significance of difference from adult.

changes in blood volume, even during periods of rest (Fig. 1a, b), despite a lack of stimulus-evoked changes. We examined differences in the power spectrum of blood volume changes during periods of behavioral quiescence across ages (Fig. 3a, b). We plotted the power spectrum on a log-log plot, and saw that power decreased roughly linearly with increasing frequency (indicating power-law-like or 1/f-like behavior[42,48]). The younger the mice were, the more pronounced the low-frequency oscillations. We fit the power spectrum of the spontaneous CBV fluctuation with a power law of the form $P = af^{-\beta}$, as is seen with tissue oxygenation and BOLD signals[42,48]. The larger the magnitude of $\beta$, the steeper the drop-off in power with frequency and the larger the contribution of low-frequency oscillations to higher-frequency ones. We found that the fitted exponent of the power-law model became less negative with age (P10: 1.846 ± 0.105, $P < 1.07 \times 10^{-7}$ versus adult; P15: 1.780 ± 0.067, $P < 1.24 \times 10^{-6}$; P20: 1.512 ± 0.118, $P < 0.0056$; P25: 1.161 ± 0.078, $P < 0.417$; P30: 1.183 ± 0.104, $P < 0.55$; Adult: 1.235 ± 0.108) (Fig. 3c), indicating that the relative contribution of low-frequency power decreases with age. We then removed the 1/f-like trend by pre-whitening to emphasize any peaks in spectral power, as there is an additional, non-neuronally driven component of vascular oscillations at ~0.1–0.3 Hz in primates and rodents[21,42,48,49]. We found that the amplitude of the peak in this range increased with age (absolute peak spectral power: P10: 0.046 ± 0.02, $P < 8.51 \times 10^{-5}$; P15: 0.046 ± 0.006, $P < 2.6 \times 10^{-4}$; P20: 0.088 ± 0.031, $P < 0.023$; P25: 0.069 ± 0.016, $P < 0.005$; P30: 0.249 ± 0.08, $P < 0.002$, Adult: 0.155 ± 0.003) (Fig. 3d), noting that absolute spectral power increases with age. When we normalized total power across the spectral range, we found that the peak power at each age was a

similar fraction of the total power (P10: 0.007 ± 0.003, $P < 0.458$; P15: 0.005 ± 5.8 × 10^{-4}, $P < 0.869$; P20: 0.006 ± 3.6 × 10^{-4}, $P < 0.920$; P25: 0.004 ± 1.0 × 10^{-4}, $P < 0.351$; P30: 0.007 ± 0.001, $P < 0.0283$; Adult: 0.005 ± 7.2 × 10^{-4}) for all animals older than P10. However, the frequency underwent a marked shift with age (Fig. 3e), where the peak in the whitened power spectrum of the spontaneous CBV at P10 and P15 was a substantially lower frequency than the peak at other ages (P10: 0.018 ± 0.003 Hz, $P < 3.6 \times 10^{-4}$, P15: 0.067 ± 0.033 Hz, $P < 0.007$; P20: 0.159 ± 0.008 Hz, $P < 0.684$; P25: 0.247 Hz±0.1, 0.088; P30: 0.198 ± 0.017 Hz, $P < 0.56$, Adult: 0.176 ± 0.003 Hz). These results suggest that there are qualitatively different vascular dynamics in the neonatal mouse than the adult. Intriguingly if these low-frequency oscillations interact with sensory stimulation, they could greatly impact the hemodynamic response[50] (see "Discussion").

**Neonatal mice show frequent sleep–wake transitions.** In adult mice, cortical blood volume is substantially higher during both NREM and REM than in the awake state, and the dynamics of these blood volume changes track sleep–wake transitions that take place on the time scale of minutes (~0.01–0.02 Hz)[29,30]. As large blood volume changes occur during sleep–wake transitions, we asked if neonatal mice were undergoing these changes, as neonatal rodents are known to sleep more than adults[51].

To determine how arousal-state transitions related to hemodynamic changes in neonatal mice, in a separate set of mice ($n = 5$ mice at P10, $n = 4$ at P15) we monitored blood volume while simultaneously measuring the nuchal muscle electromyography (EMG) signal and hippocampal LFP (Fig. 4a). We

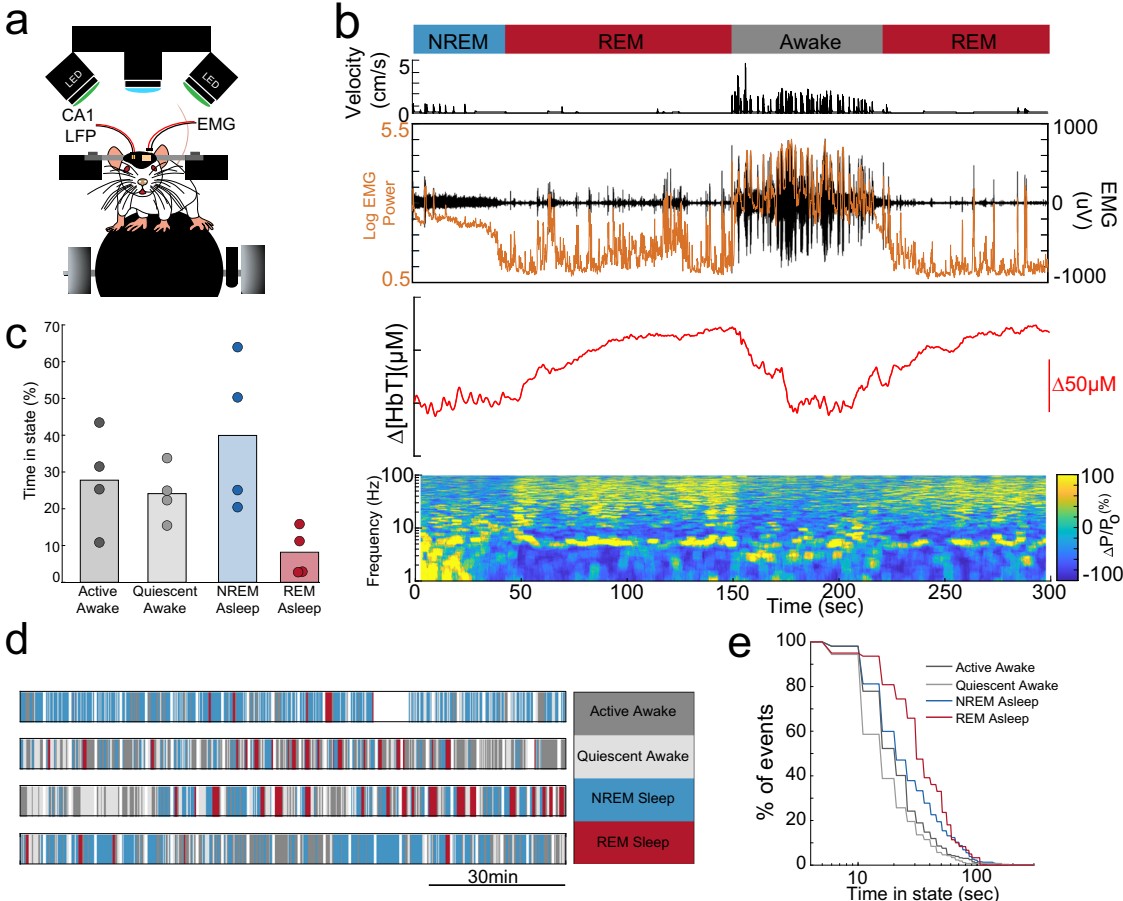

**Fig. 4 Arousal-state changes in neonatal mice. a** Schematic of the setup for monitoring arousal state and vascular dynamics. Nuchal EMG and hippocampal LFP are recorded. **b** Example of a P15 mouse transitioning through NREM, REM and awake state. Top: Arousal state, Middle: Velocity, Nuchal muscle EMG and EMG power, Bottom: Blood volume in primary somatosensory cortex, LFP spectrogram of hippocampus LFP. Note sustained motor theta and gamma-band power during REM, but not intervening periods of movement. **c** Histogram of time in each state. **d** Hypnograms for four different P15 mice. White denotes gaps in data recording. **e** Survival plots of event duration by event type. REM events are the fewest in number but longest in duration, while periods of awake quiescence are on average exceptionally brief.

manually scored arousal state in 5-s bins using nuchal muscle tone and hippocampal LFP as indicators of arousal, but were blinded to blood volume changes. At age P10, mice spent a substantial amount of time sleeping. As the sleep dynamics in rodents at P10 are not fully adult-like[51–54], we categorized the arousal states as active/quiescent asleep, or as active/quiescent-awake. We found that at age P10, the mice spent a majority of their time ($57.19 \pm 11.54\%$) in active sleep states, characterized by muscle atonia and myoclonic twitching (Supplementary Fig. 5). The active sleep state also represented the state with the longest bout duration during this time period (50% behaviors longer than $53 \pm 32$ s). While periods of prolonged behavior were observed, many behavior events occurred on time scales of shorter than a minute in duration (percent of behavior $25 \pm 5\%$, 50% behaviors less than $19 \pm 5.7$ s) (Supplementary Fig. 5).

In P15 mice, arousal state was scored as active or quiescent-awake, NREM or REM. We saw canonical hippocampal LFP markers of NREM and REM states (increased delta and theta-band power, respectively) as well as atonia in P15 mice (Fig. 4b). During REM in P15 mice, we observed large increases in total blood volume, similar to what is seen in adults. P15 mice spent $8.17 \pm 65\%$ of the time in REM, and 50% of REM events were longer than $34 \pm 13.04$ s. The NREM phase of sleep composed $39.9 \pm 20\%$ of the time and 50% NREM bouts were longer than $24 \pm 2.74$ s. Both NREM and REM events represented the longest behavioral bouts in

duration and frequent transitions between NREM and awake behavior are observed (Fig. 4c, d). P15 mice spent $27.78 \pm 4.18\%$ of the time in active behavior, with 50% of behaviors of longer duration than: $21 \pm 3.53$ s) (Fig. 4c–e), while periods of quiescent-awake occurred $24.12 \pm 4.18\%$ of the time, with 50% behaviors of longer duration than $17 \pm 4.18$ s (Fig. 4e). During REM, we observed elevated blood volume (Fig. 4b) at both ages.

These results show that under our imaging conditions, neonatal mice spent most of their time asleep and underwent arousal-state transitions on the time scale of tens of seconds. In adults, these arousal transitions are associated with large changes in cerebral blood volume, and the more frequent events in younger mice could underly the higher power at lower frequencies in neonatal mice.

**Arousal-state transitions drive large changes in blood volume.** We then asked if the state transitions were associated with large changes in blood volume Δ[HbT], as is seen with arousal-state transitions in adult mice. We visualized peri-arousal-state changes in EMG, LFP, and Δ[HbT] (Figs. 4b and 5a–d and Supplementary Fig. 6). At P15, mice often began moving upon awakening, so there were frequent transitions from NREM/REM to active awake (Fig. 4b). Transitions from awake to NREM or NREM to REM were followed by increases in blood volume ($12.35 \pm 9.41\,\mu M$, $P < 0.056$, and $12.18 \pm 8.44\,\mu M$, $P < 0.045$, respectively, mean

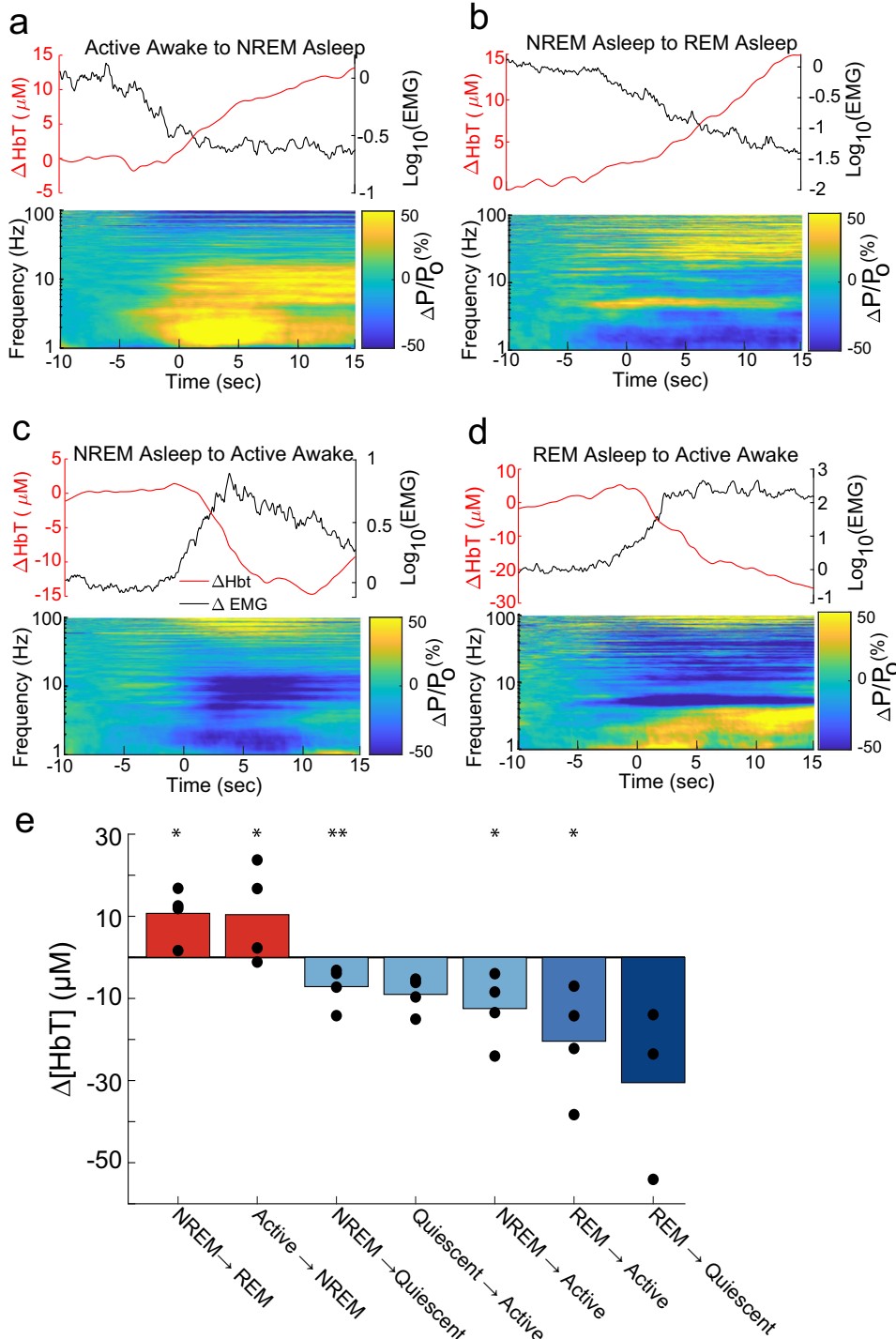

**Fig. 5 Arousal-state transitions drive large hemodynamic changes in P15 mice. a–d** Peri-transition EMG, hippocampal LFP, and Δ[HbT] are plotted relative to the arousal-state transition time for the denoted state transitions. Wake->sleep transitions drive large increases in Δ[HbT], sleep->wake transitions drive large decreases in Δ[HbT]. **e** Bar plot showing means of each transition. *$P < 0.05$; **$P < 0.01$; ***$P < 0.001$ significant difference from zero.

difference between 5 s before an arousal-state change with average of 5 s to 10 s after the state change). The transitions between quiescent-awake and NREM were not as large in magnitude, partially due to an observed oscillation in blood volume around NREM onset. Awakening from sleep drives large decreases in blood volume (Fig. 5e) (NREM to active awake: $-13.43\,\mu M \pm 7.41$, $P < 0.025$; REM to active awake: $-26.26 \pm 12.51\,\mu M$, $P < 0.017$; NREM to quiescent-awake: $-11.51 \pm 4.48\,\mu M$, $P < 0.01$; REM to quiescent-awake: $-29.14 \pm 27.98\,\mu M$, $P < 0.158$; quiescent-awake

to active awake: $-7.13 \pm 5.03\,\mu M$, $P < 0.065$). Some arousal-state transitions were rarely observed, such as REM to quiescent awakening from REM was invariably followed by transition to active behaviors. The changes in blood volume produced by awakening were often larger than those elicited by sensory stimulation (~15 μM peak changes in adults, Fig. 1e, f).

In P10 mice, transitions from periods of behavior or quiescence to periods of sleep are accompanied by increases in blood volume similar to those seen in P15 mice (Supplementary Fig. 7) (active

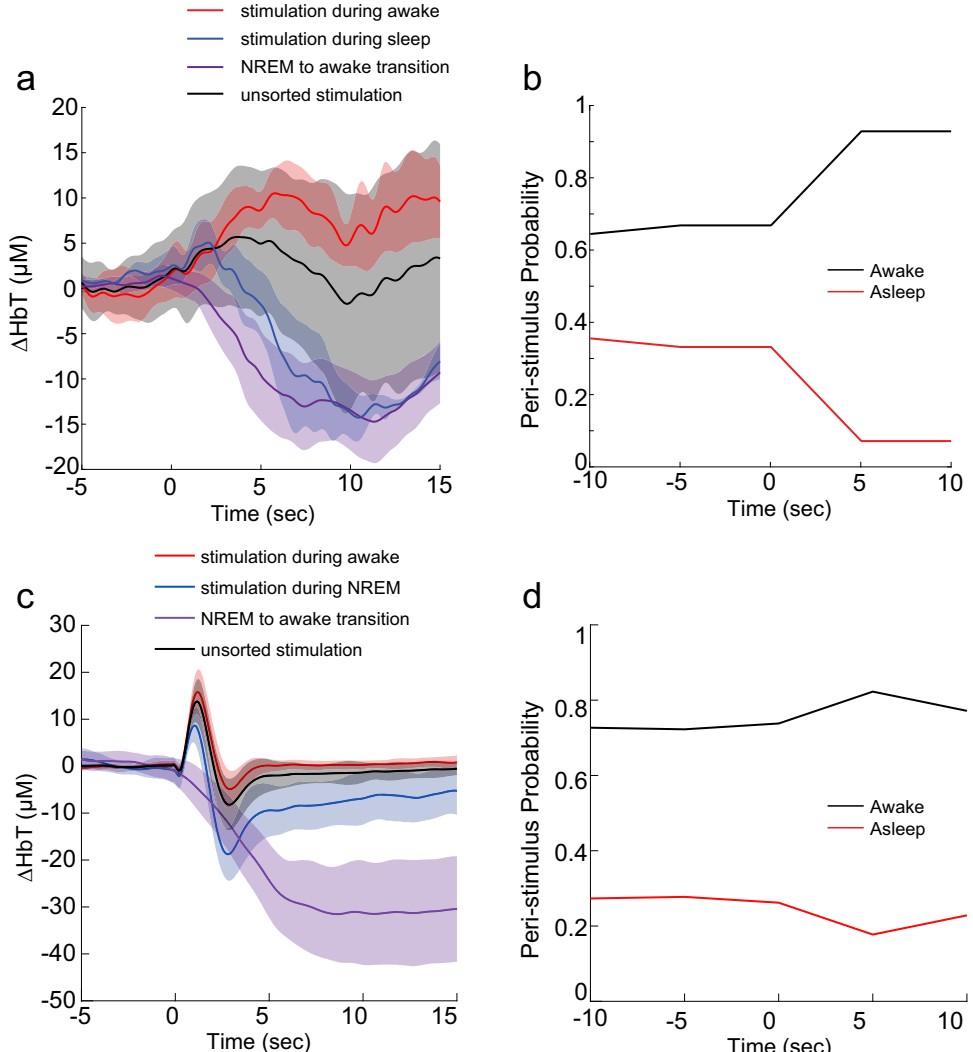

**Fig. 6 Sensory stimulation drives arousal changes that impact blood volume. a** Average stimulus-evoked Δ[HbT] responses to whisker stimulation in P15 mice separated out by arousal state, along with replotting of average NREM->awake transition. Shaded bars show standard deviation. Stimulation in the awake condition leads to dilation, stimulation in the sleeping state leads to constriction. Note that sensory stimulation during NREM is very similar to NREM->awake condition. Average of all stimuli (regardless of conditions) is weighted average sensory stimulation in awake and NREM sleep. REM is excluded from these plots as the mice spent essentially no time in this state. **b** Peri-stimulation probability of being awake and asleep in P15 mice. Stimulation causes a substantial increase in awakening. **c** Same as (**a**), except in adult mice. **d** Same as in (**b**), except for adult mice.

awake to quiescent asleep: $7.33 \pm 11.144 \, \mu M$, $P < 0.177$, quiescent-awake to quiescent asleep: $16.13 \pm 15.33 \, \mu M$, $P < 0.09$, quiescent-awake to active asleep: $16.91 \pm 4.24 \, \mu M$, $P < 5 \times 10^{-5}$ active awake to active asleep: $9.88 \pm 7.20 \, \mu M$, $P < 0.05$) and transitions from quiescent sleep to active sleep further increase blood volume ($7.03 \pm 9.41 \, \mu M$, $P < 0.13$). At P10, waking from sleep generated decreased cortical blood volume alongside increasing nuchal muscle tone (active asleep to active awake: $-23.16 \pm 4.51 \, \mu M$, $P < 2.1 \times 10^{-4}$, quiescent asleep to active awake: $-15.24 \pm 12.67 \, \mu M$, $P < 0.07$, active asleep to quiescent-awake: $-7.91 \pm 10.754 \, \mu M$, $P < 0.19$) (Supplementary Figs. 4 and 6). Many of these state transitions were rarely observed, such as active awake to quiescent asleep, and were primarily brief transition periods between active awake behaviors and active sleep. These results suggest that both before and after eye opening, changes in arousal state have profound effects on cortical blood volume levels, with awakening being associated with large decrease in blood volume. These arousal-state changes were larger and longer lasting than sensory-evoked signals.

**Stimulation drives changes of arousal state in juvenile and adult animals.** If transitioning from sleep to awake can cause a large decrease in blood volume, a stimulation that awakens the mouse could mask neurovascular coupling. To test this possibility, we stimulated the whiskers (with interstimulus intervals >1 min, jittered to prevent entrainment) while simultaneously monitoring arousal state and the hemodynamic response in P15 mice. We broke out responses that were presented in the awake state from those that were occurred when the mouse was asleep. As whisker stimulation was done early in the imaging procedure, there was little to no REM sleep, so we disregard that transition in our further analysis. We plotted the stimulus-evoked response, either averaged without regard to arousal state, split out by arousal state and the average changes elicited by the transition from NREM to awake (Fig. 6a). Whisker stimulation drove robust increases in blood volume when the mice were awake, reminiscent of neurovascular coupling in adults, whereas the same stimulus presented in the sleeping state caused a decrease in blood volume that was very similar to that that occurred during

awakening. When we look at the arousal state before and after the stimulus, we found that P15 mice were awakened by the stimulation, going from being awake ~65% of the time to ~95% (Fig. 6b). We reanalyzed a preexisting data set of adult mice[29] that had their whiskers simulated, and that occasionally were in NREM sleep during the stimulation (Fig. 6c, d). Stimulation during sleep elicited an increase in blood volume that was smaller than stimulation during the awake state, and that was followed by a sustained decrease in blood volume. However, the adult mice were mostly awake, and stimulation did not cause as much awakening as in the P15 mice. These results show that sensory stimulation in awake neonates drives a positive blood volume response, but in sleeping neonates it triggers an awakening that swamps any positive response.

## Discussion

Using optical measures of cortical blood volume and fMRI BOLD signals, we have shown that in head-fixed neonatal mice the hemodynamic response to somatosensory stimulation differs from that of adult animals, with sensory-evoked increases in blood volume in neonates being smaller than those in adult animals, and in some cases followed by a pronounced decrease in blood volume. The same age-dependent pattern in BOLD responses was observed in both somatosensory and visual systems in restrained mice. These differences cannot be attributed to differences in neural activity, as by P15, neural responses in the somatosensory cortex to sensory stimulation were comparable to those in adults. This is consistent with previous investigations of whisker barrel cortex, which have shown excitatory and inhibitory neurons of layer 4 receive strong excitatory input from thalamus[55–57] and that whisker stimulus excites primary sensory cortex within the first postnatal week[44,58]. Relative to adults, neonatal mice display much greater power in blood volume oscillations at very low frequencies (~0.01 Hz) that are associated with sleep[29,30], and much lower power in the "vasomotion" frequencies (~0.1–0.3 Hz) that are independent of neural activity[21,42] (Fig. 3). The slope of the power law fit became smaller as the animals aged, reaching adult-like spectral distribution in the fourth postnatal week.

There are several caveats to this work. Though the microvasculature is still developing in neonatal mice, this process is largely complete by P15[5,8], so it is not likely to account for the differences in responses between young mice and adults. While we monitored locomotion behavior of mice during imaging, small movements such as whisking, and fidgeting were not tracked, though these movements are highly correlated with the arousal state, which we did track. Fidgeting behaviors are known to drive increases in neural activity and blood volume in adults[21,23,59], and active whisking and exploration develop rapidly in the third postnatal week[60,61]. However as somatosensory stimuli fail to evoke large increases in blood volume, it is assumed fidgeting and volitional whisking would not alter this relationship. As fidgeting movements are short-duration events, their hemodynamic responses should be equally brief and occur on the order of seconds. The power spectra of blood volume changes in young animals show a lack of power within these behaviorally relevant bands, suggesting that either few of these events were interspersed within our collected data or, these events fail to elicit changes in hemodynamics in an adult-like manner. We measured changes in hemoglobin concentration, not direct change of vascular diameter or blood flow, and these reflective signals are a mix of arterial, venous, and capillary components[38]. In previous work in adult mice, it has been shown that even though the venous system has a larger volume, arterial volume changes dominate changes in the optical signal[38,62], because arterial diameter changes are so much larger than those of veins[63,64]. However, we do not know if this holds for arousal transitions in neonates. Awakening is also accompanied by changes in heart rate and respiration[29] which could also contribute to the changes we see here. In the awake, unanesthetized adult mouse, pharmacological manipulations of blood pressure have little effect on blood flow and arterial changes[62] due to autoregulation. In adults, the frontal areas show no increase in blood flow or volume during locomotion[22,24,37,62], and local silencing of neural activity in somatosensory cortex blocks arterial dilation during locomotion[65]. These results show that vascular responses in the adult cortex are buffered from systemic changes by autoregulation and are under local control. However, autoregulation is not thought to be as effective in neonates[66]. If autoregulation is not as effective in neonates, then increases in blood pressure during awakening will tend to counteract the increase in resistance due to vascular constriction, resulting in a smaller decrease volumetric flow than if autoregulation were fully effective. Respiration rate changes accompanying arousal-state changes will alter blood gasses and pH, but these changes are probably too slow to explain the rapid constrictions we see here, as inhalation of 5% $CO_2$ takes minutes to drive appreciable changes in arterial diameter[67]. We did not measure body temperature, and the neonates likely had lowered body temperature during imaging. Cooling of the adult brain below ~20 °C results in large reduction of neural activity and a profound decrease in the hemodynamic response[68], but neonates separated from their mother typically maintain a temperature above 30 °C[69]. Circadian rhythms can also affect temperature, but brain temperature in adults changes only by a few tenths of a degree over minutes[70], so it is hard to attribute the rapid constrictions upon awakening seen here to any temperature change. However, cooling of a few degrees in adults causes a slowing of the hemodynamic repsonse[68], and may account for the slower dynamics of neonates compared to adults.

Consistent with previous reports, we found that sensory stimulation in neonatal mice and rats drove a small positive response, followed by a delayed and large decrease in blood volume[9–11]. Also consistent with previous reports, our BOLD fMRI experiments showed a smaller BOLD response in younger mice[15]. However, monitoring of the arousal state showed that neonatal mice spend a substantial amount of time asleep. When stimulation was done when the mice were asleep, they transitioned to the awake state, and the delayed decrease in blood volume could be attributed to the vasoconstriction associated with awakening[29,30], not inversion of neurovascular coupling. In neonatal rats at age P5, spontaneous neural activity is higher during active sleep than the awake state[71,72], though spontaneous movements increase activity relative to non-movement periods in the awake state. Thus, awakening could drive an overall decrease in neural activity.

Both head-fixed[29,30,73] and freely moving[74] mice frequently sleep with their eyes open. This may explain the lack of visual response in V1 at young ages (Fig. 2). Relating our results to previous developmental studies with anesthetized animals is more complicated. In adults, sleep is associated not only with large changes in local neural activity, but also neuromodulation changes[75], and increases in noradrenergic signaling is associated both with awakening[76] and vasoconstriction[77,78]. There is no single anesthetized state[79], and different anesthetics have varying effects on neurons and the vasculature[80], with some anesthetics giving more sleep-like activity than others[81]. For example, there are similar arousal transitions during urethane as during normal NREM sleep[82]. If stimulation causes animals to transition to a more awake-like, vasoconstricted state (with higher noradrenergic tone) this would explain the similarities between our observations and those made in urethane anesthetized animals[9,11].

When P15 or adult mice were stimulated in the awake state, there was an unambiguous increase in blood volume (Fig. 6). The vasoconstriction driven by awaking is likely due to the increase in noradrenergic tone (which is vasoconstrictive[77,78]) that accompanies awakening[30,76,83–85]. Our work suggests that the disparate observations of the hemodynamic response in neonates in previous studies may be caused by differences in baseline arousal level of the subjects (due to differences in illumination, scanner noise etc.). These results are also in line with a recent study in unanesthetized rats, which showed that a brain-wide negative BOLD signal that could dominate over positive, sensory-evoked BOLD responses was observed in approximately one-third of all fMRI trials[50]. In these negative BOLD trials, neural responses remained positive, evidenced by consistent response of the simultaneously recorded calcium signal. This distributed negative BOLD signal is likely also due to the awakening of animals in these trials, although this possibility needs to be further validated with additional measurements. In human adults, awakening from sleep drives large decreases in cortical BOLD[86], so we suspect a similar process is taking place in humans.

Hemodynamics responses are not driven only by sensory-evoked activity, they are also strongly affected by changes in arousal, and the hemodynamic decreases caused by awakening can overwhelm those caused by sensory stimulation. As neonatal mice and other mammals frequently and easily cycle between wake and sleep states, their cerebrovascular responses to sensory stimulation will be dominated by arousal transitions.

## Methods

**Ethical approval**. All experiments were done in accordance with institutional approval by the Pennsylvania State University Institutional animal care and use committee (IACUC) in accordance with NIH guidelines.

**Statistics and reproducibility**. Animal numbers were chosen to be consistent with previous studies on the physiology of neonatal rodents[5,31]. Unless otherwise indicated, all statistics were done using generalize linear models (GLMs) using the animal identity as a random effect and the arousal-state transition as a fixed effect. GLMs account for within-animal correlations and correct for multiple comparisons[34,35]. A total of 80 and 11 Swiss webster mice (45 male) were used in the optical imaging and electrophysiology experiments, respectively. In total, 49 Swiss webster mice were used in the fMRI experiments (whisker stimulation: $n = 22$, visual stimulation: $n = 27$). Ages were chosen to span the period of rapid changes around eye opening (~P14) in which neural responses in the barrel cortex become more adult-like[47,60] and to detect any more subtle changes in the following weeks.

To avoid confounds of due to potential disruption of growth, all optical imaging/electrophysiology experiments were acute. For fMRI experiments, all mice were imaged longitudinally. However, data from some imaging points were not included due to excessive motion during imaging or animal mortality. In the visual stimulation experiment ($n = 38$), fMRI data used in the final analysis include: five mice imaged for all six time points, six mice imaged for five time points, four mice imaged for four time points, three mice imaged for three time points, twelve mice imaged for two time points, and eight mice imaged one time point. In the whisker stimulation experiment ($n = 26$), fMRI data used in the final analysis include: five mice imaged for all six time points, eight mice imaged for five time points, two mice imaged for four time points, four mice imaged for three time points, five mice imaged for two time points, and two mice imaged for one time point.

**Surgical procedures**. A polished and re-enforced thin skull window was made over right parietal cortex for IOS imaging of cortical vasculature[21,29,37,65,87]. Briefly, animals were initially anesthetized with 5% isoflurane in oxygen and maintained with 2% isoflurane in oxygen for the duration of the procedure. The scalp was resected, and the periosteum gently removed from the skull. A titanium head bar was fixed to the skull using cyanoacrylic glue and dental cement. The skull was thinned using a pedal-controlled dental drill (Foredom) and #005 conical dental burrs. The skull was then polished using 3 F dental abrasive, washed with sterile water and then allowed to dry. A #0 coverslip was then attached over the window using cyanoacrylic glue. For experiments where arousal state was monitored, a small craniotomy was made contralateral to the window in parietal cortex (~2.5 mm caudal from Bregma, +2 mm lateral from the midline). A custom-made tungsten wire (#795500, A-M Systems, Sequim, WA) stereotrode ensheathed in polyamide tubing (#822200, A-M Systems, Sequim, WA) was implanted vertically into the CA1 region of hippocampus (~1–1.5 mm below cortical surface depending

on animal age). The craniotomy was sealed with KwikSil silicone (WPI) before the stereotrode was anchored to the head-bar apparatus with dental cement. For electromyography (EMG) stereotrodes, the skin of the neck was resected exposing the nuchal muscles at the base of the skull. A custom-made braided stainless-steel wire (#793200, A-M systems, Sequim, WA) stereotrode ensheathed in polyamide tubing (#822200, A-M Systems, Sequim, WA) was implanted into the outer most left nuchal muscle for EMG recording. The implant site was sealed using KwikSil and then anchored to the head bar using dental cement. Following completion of surgery, animals were removed from isoflurane and placed in a temperature-regulated incubator to maintain core body temperature during recovery from surgery.

**Craniotomy for laminar electrophysiology**. Animals were anesthetized as above. The scalp was removed, and a titanium head bar was anchored to the skull using cyanoacrylic glue and dental cement. A craniotomy was made over whisker barrel cortex and the dura was resected, exposing the cortical surface. The brain surface was kept moist using Vetspon soaked in ACSF and the craniotomy covered with Kwik-Cast. Following surgery, animals were allowed to recover in an incubator for ~2 h prior to head fixation and recording.

**Animal habituation**. For imaging experiments in mice without hippocampal electrodes, animals were allowed to recover for 24 h before the first head fixation. Due to the rapid growth during the early postnatal period, only a single 30 min habituation session was performed for mice implanted with electrodes were imaged. Animals were head-fixed on a spherical treadmill and placed in the IOS imaging area the day prior to data collection for 30 min before being returned to their home cage. For arousal-state-based scoring experiments, animals were allowed to recover post-operatively for 3 h followed by 30 min of head fixation prior to imaging experiments on the same day as the surgery. Treadmill movement was recorded using a rotary encoder[38].

**Intrinsic optical signal imaging**. The dorsal surface of parietal cortex was illuminated with four 530 nm LEDs (Thor Labs 530L3) that were bandpass filtered at $530 \pm 10$ nm (Thor Labs FB530 ± 10 nm). Changes in reflected light at 530 nm (an isobestic point of hemoglobin) report blood volume changes[38,39,88]. Images (256 × 256 pixel, 12-bit depth) were acquired at 30 frames per second with exposure duration of 20 ms using a Dalsa 1M60 camera fitted with a Thor Labs FEL500 500-nm long-pass filter. Animals were imaged in sequential 5-min trials for a total duration of 2 h. Image acquisition was synchronized with sensory stimulus presentation and analog signal acquisition in custom-written National Instruments LabView software.

**fMRI experiments**. fMRI experiments were conducted on a 7 T MRI system interfaced with a Bruker console (Billerica, MA) at the High Field MRI Facility at the Pennsylvania State University. Gradient-echo images were acquired using echo-planar imaging (EPI) with the parameters: repetition time (TR) = 1 s; echo time (TE) = 15 ms; matrix size = 64 × 64; FOV = 1.6 × 1.6 cm$^2$; slice number = 16; and slice thickness = 0.75 mm, flip angel = 60°.

In the visual fMRI experiments, blue light stimulation was generated by an LED and delivered by an optical fiber. Light power was maintained at 0.5 mW/mm$^2$ at the optical fiber tip, which was placed about 5 mm in front of the left eye. Ten epochs of stimulation were repeated in each fMRI scan. In each epoch, light with different frequencies of 2, 5, and 10 Hz at 50% duty cycle was turned on for 2 s, and followed by 28 s of darkness. Each animal was scanned for six times with two scans for each frequency. For whisker stimulation, we used air puffs to deflect the left-side whiskers. Before each fMRI session, whiskers on the right side were trimmed to 1 cm. The air puff was kept at 6 psi, and was delivered at 12 Hz to achieve maximum BOLD response[89]. Ten epochs of stimulation were repeated in each fMRI scan, with a 5-s on and 25-s off period for each epoch. For both visual and whisker fMRI scans, the first epoch was preceded with a 25-s baseline scan.

EPI images for different age groups were separately registered to the corresponding developing mouse brain templates with the most closely matched age (P10 - > P7, P14/15 - > P14, P20 - > P21, P25/30 - > P28, P60 - > P63) obtained from the Center of Magnetic Resonance Microimaging at John Hopkins Medical Institute (http://lbam.med.jhmi.edu). These templates were further nonlinearly registered to the Allen Mouse Brain Atlas (http://www.brain-map.org/)[90] using ANTs software (Advanced Normalization Tools, http://picsl.upenn.edu/software/ants/)[91] for the definition of anatomical regions of interest (ROIs: visual cortex, superior colliculus, somatosensory cortex).

**Electrophysiological recording**. Following completion of craniotomy and recovery, animals were removed from anesthesia, transferred, and head-fixed to a spherical treadmill and placed in the electrophysiology recording apparatus. A single shank multi-site probe from Neuronexus was implanted parallel to the cortical surface targeting the most lateral whisker barrels, rows D–E, until the topmost channel reached the surface of the brain. Following probe implantation, animals were allowed to recover for ~30 min, or until visible multiunit spiking was observed in the recording data. Data was collected for approximately 2 h from each animal while animals were able to freely locomote. Broadband (0–10 kHz)

waveforms were recorded using Neuronexus acquisition software and digitized at 20 kHz.

**Whisker stimulation**. Stimulation of contralateral and ipsilateral whisker pads and a control stimulus directed away from the animal consisted of a 0.5 s pulse of air (10 psi) delivered to the E row of the facial vibrissae. Stimulus order was randomized within 5 min trials with 60 s interstimulus intervals.

**Histological reconstruction of the imaging region**. Following the completion of data acquisition, animals were euthanized by transcardial perfusion first with heparinized saline, followed by 4% paraformaldehyde (PFA). Fiduciary marks were made at the corners of the thin skull window through the brain to position imaging location relative to primary whisker barrel cortex. Brains were removed from the skull and cryo-protected by storage in a 4%/30% PFA/sucrose solution until the brain had equilibrated to the solution ~24 h. The imaged cortical hemisphere was removed, flattened, and 80 μm tangential tissue sections were cut along the dorsal-ventral axis of cortex. Whisker barrel cortex was visualized using cytochrome-oxidase labeling, and sections were assembled using Adobe Illustrator to reconstruct the somatosensory cortex under the window[21,34].

**Calculation of Δ[HbT]**. Changes in image pixel intensity are inversely related to the concentration of oxygen-bound and unbound hemoglobin (Δ[HbT]). Following the histological reconstruction of the somatosensory cortex, a region of interest was drawn over the pixels corresponding to the primary whisker barrel cortex. Pixel intensities inside the ROI were averaged to generate the ROI intensity value. Calculation of Δ[HbT] was performed using the modified Beer–Lambert law:[39]

$$\Delta[HbT(t,\lambda)] = \frac{-\frac{1}{X(\lambda)} * \ln\left(\frac{I_{(t,\lambda)}}{I_{(t_0,\lambda)}}\right)}{\xi(\lambda)}$$

Where I is the recorded intensity, $I_0$ is the intensity of light incident on to the pial surface, $\mu_a$ is the absorption coefficient of the brain, X is a wavelength-dependent estimation of pathlength differences between incident photons due to scattering within tissue and $\xi(\lambda)$ is the wavelength-dependent molecular extinction coefficient HbO and HbR.

**Power-law fitting of resting-state hemodynamics**. A subset of animals was allowed to freely behave on the spherical treadmill free of whisker stimulation. Trials where movement was detected for less than 10% of the trial duration were used to assay the "resting-state" dynamics of Δ[HbT]. Fitting of power spectra with was performed as described in ref. [42]. Briefly, the power spectrum was first resampled using logarithmically spaced frequency bins to account for uneven sampling of lower frequency bins when converted to a logarithmic scale. Following resampling, the logarithmic power spectrum was fit with a power-law distribution using ordinary least squares regression.

**Analysis of electrophysiology data**. Broadband electrophysiology signals were bandpass filtered into multiunit activity [300 Hz–5 kHz] and LFP [1–100 Hz] bands using zero-lag filtering (MATLAB: filtfilt). For multi-channel array cortical recordings, spikes were identified as voltage changes >4 standard deviations above or below the mean for each channel, with a minimum 1 ms refractory period applied between detected threshold crossings. Spike rates were estimated by convolving binarized spike trains with a normalized Gaussian with a standard deviation of 100 ms. The frequency band-dependent spectral power of LFP was estimated using multi-taper analysis using the Chronux toolbox for MATLAB (mtspecgramc)[92]. Channels corresponding to the cortical layers within whisker barrels were identified using the current source density method by calculating the second spatial derivative of LFP waveforms and identifying current sinks associated with stimulus-evoked thalamic input to layer 4. The hippocampal LFP were used for manual scoring of animal arousal state. Broadband signals were bandpass filtered between 1 and 150 Hz using zero-lag filtering (MATLAB: filtfilt). LFP power spectrograms were calculated for entire 5 min data using a multi-taper estimate method (MATLAB Chronux toolbox: mtspecgramc). A sliding window 5 s in length with 0.5 s steps was used to estimate spectral power within the window. Spectrograms for all data sets were normalized against spectral power calculated from at least 2 min of rest identified within the first 20 min of an imaging session. Nuchal muscle EMG signals were bandpass filtered between 300 Hz and 5 kHz, rectified, squared and then convolved with a Gaussian kernel with a standard deviation of 50 ms.

**Arousal-state scoring**. All arousal-state scoring was done blind to the Δ[HbT] signal. All acquired data was manually scored with 5 s resolution. Due to the developmental differences between P10 and P15 mice, we used two different arousal scoring systems. For P10 mice, arousal states were classified into three behavioral categories, active awake, active asleep, and behavioral quiescence. Active awake was characterized by high nuchal muscle EMG tone and active locomotion. Because increased delta band power in recorded LFPs during NREM sleep emerges around P11, there is limited ability to differentiate between periods of behavioral quiescence and NREM sleep[32,51,93]. Periods of behavioral quiescence were defined

as periods excluded from active awake or 3. Active asleep. P15 mice were classified into active awake, quiescent-awake, NREM, and REM. Active awake had loco-motion, high EMG power, and increased hippocampal theta and gamma-band power relative to quiescent-awake. Quiescent-awake was defined as behavioral quiescence without muscle atonia. NREM sleep was defined as behavioral quiescence, EMG atonia, elevated delta band LFP power. REM sleep defined as, muscle atonia with interspersed myoclonic twitching, and elevated hippocampus theta and gamma power.

**Analysis of arousal-state transitions**. Events for arousal-state transition averaging had a minimum of 15 s within the preceding state followed by a minimum of 15 s duration in the subsequent state. Baseline [HbT] values were calculated as the mean [HbT] for 15 to 5 before a change in arousal state. Evoked [HbT] changes were calculated as the difference from this baseline, and average state evoked changes in [HbT] were calculated as the mean [HbT] from 5 to 15 s following a change in arousal state.

**Reporting summary**. Further information on research design is available in the Nature Portfolio Reporting Summary linked to this article.

## Data availability
Data from the figures for this study are available on Dryad (https://doi.org/10.5061/dryad.k6djh9wbt). Data for the figures are also available as Supplementary Data 1.

## Code availability
Code to generate the figures for this study is available on Zenodo (https://doi.org/10.5281/zenodo.7809100)[94].

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

## Acknowledgements

P.J.D. and N.Z. are supported by Grant R01NS101353 from the NIH; P.J.D. is supported by R01NS078168 and U19NS128613. Q.Z. is supported by a Career Development Award (#935961) from the American Heart Association. H.S.Ü. is supported by a Republic of Türkiye Ministry of National Education Scholarship.

## Author contributions

Conceived and designed experiments: K.W.G., N.Z., and P.J.D.; performed experiments: K.W.G., H.S.U., X.H., Q.Z., and K.L.T.; analyzed the data: K.W.G., H.S.U., X.H., Q.Z., and K.L.T.; supervised experiments: N.Z. and P.J.D. Wrote paper: K.W.G. and P.J.D. with input from all authors.

## Competing interests

The authors declare no competing interests.
