## [Peer Review File · Communications Biology]

Reviewers' comments:

Reviewer #1 (Remarks to the Author):

This is an interesting manuscript using a multidisciplinary approach to investigate the coupling between neural activity and cerebral perfusion in awake mice during the post-natal period. It is reported that at P10 somatosensory stimuli evoke neural activity in the mouse neocortex, but the corresponding hemodynamic response is almost absent. The magnitude of the hemodynamic response increases over time eventually reaching levels comparable to adulthood (>P30). Monitoring of the sleep wake cycle, demonstrated that P10 mice cycle frequently between sleep and wakefulness and that the transient reduction in blood volume associated with arousal, reflecting vasoconstriction, presumably obfuscates the vascular changes induced by activation. Later postnatally, the hemodynamic changes induced by activation become more apparent because the mice spend more time in the awake state. It is concluded that the arousal state rather than changes in neural activity may be responsible for the changes in the neurovascular occurring in the post-natal period.

This paper has several strengths. The experiments were performed in the awake state, both hemodynamic parameters and neural activity were monitored, and BOLD MRI was also performed which add to the translational relevance of the observations. However, some aspects of the study require clarification.

1. In addition to arousal, other physiological changes take place in the sleep-wake cycle that could also contribute to the findings. In particular, changes in blood pressure, body temperature and blood gasses, which have profound effect on CBF. These need to be discussed.
2. Was the temperature of the mice recorded and controlled during recordings?
3. What is the significance of the eye opening in the observed hemodynamic changes? Could changes in the global network activity influence the hemodynamic response to activation.
4. Cerebral blood flow or vascular diameter were not directly measured, this needs to be pointed out in the interpretation of the results.
5. In particular, changes in indices of blood volume are weighted towards veins, which may need mentioning
6. Why was neural activity measured in the hippocampus?
7. Figures 1E: the lines in the legend are not seen very well (also figure 3), which makes hard interpret the figures.
8. Figure 1D: unclear to what right and left whisker simulation and auditory stimulus refer to.
9. Data in the text make reading the results section cumbersome.

Reviewer #2 (Remarks to the Author):

Gheres et al. provide an important study about how arousal state during the neonatal period can interfere with neuroimaging results and should be consider during its performance. The study with clear objectives is well designed to answer them. I just have some comments that would like to be addressed to get the best out of this significant work.

Major comments:

Abstract:

1) On the abstract page 1 line 31 line should be mentioned this neonatal-specific result: 'These result show 31 that sleep-related vascular changes dominate over any sensory-evoked changes'.

Introduction:

1) At page 2 line 43 the authors stated the following sentence 'neonatal rodents (before postnatal day 20, P20)'. I understood what the authors want to say but P17-21 is consider infancy in rodents. Would be better reformulate to 'neonatal and infancy rodents'.

2) At page 2 line 49 the authors wrote the following 'As the GABA reversal potential becomes inhibitory in cortical neurons by P3'. According, with the paper referenced 'Hippocampal interneurons become inhibitory by P7'. Please correct. Moreover, in the literature: 'In female hippocampal and midbrain slices the GABA shift is several days earlier compared to males (approximately P6-10 in females and P14-17 in males) (Galanopoulou, 2008; Kyrozis et al., 2006; Nuñez and McCarthy, 2007), while the GABA shift in the cerebellum is actually advanced in males by 4 days (Roux et al., 2018). Thus, the timing of the GABA shift appears strongly dependent on cell type, sex and brain region.' Also, some GABA migrate from pial during postnatal period (Léger, et al 2020). With this said, not sure if the authors should state something is not totally proven yet for the brain region they are studying.

Methods:

1) The neonatal is a critical period for neurodevelopment. I wonder if the invasive procedures (craniotomy, electrophysiology among others) used along this study might disrupt the physiological development. Such as electrodes implantation create inflammation or disturb the vasculature development when dura is removed (P10). Did the authors evaluate the developmental milestones and weight? Could be beyond some results?

2) The authors mention many times in the manuscript adult-like but is not clear what is the age that adult-like is referring to. Please state it in the methods. Related with this question can the authors also explain the selection of ages?

3) Along the methods is not clear is the study is a longitudinal or terminal experiments. Please specify. If longitudinal, or same animals used for the same experiments would be a great addition correlations within the NVC bold results and arousal status.

Results

1) In the results section will be easier to follow if the authors describe the outcomes from the younger to older, to show the normal developmental process timeline. Namely at the 'neonatal mice show frequent sleep-wake transitions'.

2) Fascinatingly P10 neurons fired indeed, while quite differently from the other age P15 or P30, first they started in almost zero Hz then after fire there is a decrease on the spike rate. I am not familiar with neuronal activity, is this a normal response or neonatal related?

3) Supplemental Figure 1 – the figure has P30 but the legend refers P15. 'Neural responses to whisker stimulation in a P15 mouse.' Are the results for what age?

4) Regarding with the visual stimulation and visual cortex response, at P14 the animals should have the

eyes open, and I was expecting a V1 BOLD response at this age. However, the results showed a small response at SC but nothing at visual. Any possible explanation for this unexpected result?

5) If P10 spent the most of time in active sleep this should not mean the neurons would be in a more 'activated' status or not activate the NVC in the baseline?

6) The previous literature has tested a similar approach but using anesthesia animals. What stage of sleeping or arousal this will represent comparing with your results? If P10 or P15 are in a sleeping mode, the anesthesia should not have that much impact because it is causing similar vascular dilatation and the stimulation will awake the same way the animal. So, the aftermath will be the same right?

7) The authors mentioned that some of BOLD response might not be due to developmental processes but adult habituation. Maybe I understood wrongly but all the animals are exposed to the same machine acquisition noise. If the non-anesthetized P10 pup spends most of the time sleeping, I do not think the noise or habituation will have an impact on the BOLD/hemodynamics response in this study and this will be developmental. Would it not be possible to have some habituation in pups 1 or 3 days to see if this has really an impact as the authors said.

8) Supplemental Figure 5 – I believe the authors instead of 'Active asleep' on B and C wanted to mention 'REM sleep' to be consistent with the remaining results. Or at least match the colors of A.

Discussion:

1) There are some important results in the manuscript the authors could try to discuss/explain within the discussion perhaps with literature. Such as:

a) 'These results suggest that there are qualitatively different vascular dynamics in the neonatal mouse than the adult. Intriguingly if these low-frequency oscillations interact with sensory stimulation, they could greatly impact the hemodynamic response (49).'

Minor comments:

1) The Figure 1 has included 'Auditory stimulus' that was never mentioned in the results. Not sure if this was a typo at least it was not really tested perhaps it is a caveat.

2) Still in Figure 1 page 17 line 551-552 – 'Note that for the P10 age group, 4 animals lacked a clear peak and recovery in $\Delta[\text{HbT}]$ and were excluded from the FWHM and peak calculations.' Eventually could be on the methods and not on the legend. However, any explanation why this is happening?

3) Page 7 line 22 – typo in the references.

4) Figure 1 legend: 'Imaging blood volume changes in the somatosensory cortex of neonatal mice.' Since the authors have also 4 months included at this dataset it would be fairer 'Imaging blood volume developmental changes in the somatosensory cortex of mice.' Or 'Imaging blood volume changes across life-stages in the somatosensory cortex of mice.'

5) Supplemental Figure 1: What does this mean? 'Same as in B.'

6) Page 14 line 472, typo on the units 'um'.

Reviewer #3 (Remarks to the Author):

Overall: The authors propose disambiguating the (at least) 2 distinct hemodynamic responses likely triggered in barrel cortex on stimulating the whiskers of mouse neonates (age: P10, P15). Whisker stimulation evokes neural activity which leads to increased local blood volume, albeit an order of magnitude more slowly than in adults. At the same time, the stimulation likely wakes the neonates as they typically spend a majority of their time sleeping. This sleep to wake transition leads to a sharp decrease of local blood volume, modulating and reversing the stimulus-triggered hyperemia. The authors propose that this combination of two hemodynamic phenomena accounts for the biphasic stimulus-evoked hemodynamic response observed in neonates.

The evidence presented is compelling and the writing is clear. I have no comments other than a few very minor copy editing points. There are a few typos (line 60: 'awakening IS causes ...'. Line 226: 'NEM' in place of 'NREM'.) etc. Also suggest replacing line 239/240: '50% behaviors LESS THAN 53+/- 32 s' with '... longer than ...' since the authors are drawing attention to the striking length of the active sleep bouts. This is the same phrasing as when quantifying long REM epochs in P15 (line 225/226), as vs the 'less than...' phrasing used in lines 241/242 when pointing out the striking brevity of wake epochs.

We thank the reviewers for their enthusiasm for our work and their thoughtful and constructive critiques. Below we detail our responses point-by-point, reviewer comments are italicized.

Reviewers' comments:

Reviewer #1 (Remarks to the Author):

This is an interesting manuscript using a multidisciplinary approach to investigate the coupling between neural activity and cerebral perfusion in awake mice during the post-natal period. It is reported that at P10 somatosensory stimuli evoke neural activity in the mouse neocortex, but the corresponding hemodynamic response is almost absent. The magnitude of the hemodynamic response increases over time eventually reaching levels comparable to adulthood (>P30). Monitoring of the sleep wake cycle, demonstrated that P10 mice cycle frequently between sleep and wakefulness and that the transient reduction in blood volume associated with arousal, reflecting vasoconstriction, presumably obfuscates the vascular changes induced by activation. Later postnatally, the hemodynamic changes induced by activation become more apparent because the mice spend more time in the awake state. It is concluded that the arousal state rather than changes in neural activity may be responsible for the changes in the neurovascular occurring in the post-natal period.

This paper has several strengths. The experiments were performed in the awake state, both hemodynamic parameters and neural activity were monitored, and BOLD MRI was also performed which add to the translational relevance of the observations. However, some aspects of the study require clarification.

1. In addition to arousal, other physiological changes take place in the sleep-wake cycle that could also contribute to the findings. In particular, changes in blood pressure, body temperature and blood gasses, which have profound effect on CBF. These need to be discussed.

We have added discussion of these physiological variables to the Discussion section (see line 340-367).

2. Was the temperature of the mice recorded and controlled during recordings?

No, we did not measure temperature and we have added a discussion of this to the Discussion section (line 359-367).

3. What is the significance of the eye opening in the observed hemodynamic changes? Could changes in the global network activity influence the hemodynamic response to activation.

Neural responses become more adult-like around eye opening (Borgdorff et al., 2007; van der Bourg et al., 2017). We have added a statement about this at the beginning of the Methods section (line 419) where we also address Reviewer 2's question about the choice of age ranges.

4. Cerebral blood flow or vascular diameter were not directly measured, this needs to be pointed out in the interpretation of the results.

5. In particular, changes in indices of blood volume are weighted towards veins, which may need mentioning

We agree with the reviewer's points here. We have added discussion of these points, as well as the potential impact of temperature to the Discussion section, starting at line 359

6. Why was neural activity measured in the hippocampus?

We did this to sleep score. REM sleep is marked by a pronounced increase in the hippocampal theta band (Turner et al., 2020, eLife).

7. Figures 1E: the lines in the legend are not seen very well (also figure 3), which makes hard interpret the figures.

We have made the lines thicker.

8. Figure 1D: unclear to what right and left whisker stimulation and auditory stimulus refer to.

We have added the following to results section line 90:

“We stimulated the left and right whiskers with brief puffs of air, and a puffer not aimed at the mouse provided an auditory stimulus, but only the contralateral whisker stimulation was analyzed here”

9. Data in the text make reading the results section cumbersome.

We agree that this putting the data in the text alters hurt with the flow, but the alternatives are a dozen tables or putting the data in the figure legends, either of which would also impede flow.

We feel that neither of these two options is an improvement over leaving the data in the results section.

Reviewer #2 (Remarks to the Author):

Gheres et al. provide an important study about how arousal state during the neonatal period can interfere with neuroimaging results and should be consider during its performance. The study with clear objectives is well designed to answer them. I just have some comments that would like to be addressed to get the best out of this significant work.

Major comments:

Abstract:

1) On the abstract page 1 line 31 line should be mentioned this neonatal-specific result: ‘These result show 31 that sleep-related vascular changes dominate over any sensory-evoked changes’.

We show this for adult mice in figure 6, so we believe the statement is correct as it is.

Introduction:

1) At page 2 line 43 the authors stated the following sentence ‘neonatal rodents (before postnatal day 20, P20)’. I understood what the authors want to say but P17-21 is consider infancy in rodents. Would be better reformulate to ‘neonatal and infancy rodents’.

Corrected.

2) At page 2 line 49 the authors wrote the following ‘As the GABA reversal potential becomes inhibitory in cortical neurons by P3’. According, with the paper referenced ‘Hippocampal interneurons become inhibitory by P7’. Please correct. Moreover, in the literature: ‘In female hippocampal and midbrain slices the GABA shift is several days earlier compared to males (approximately P6-10 in females and P14-17 in males) (Galanopoulou, 2008; Kyrozis et al., 2006; Nuñez and McCarthy, 2007), while the GABA shift in the cerebellum is actually advanced in males by 4 days (Roux et al., 2018). Thus, the timing of the GABA shift appears strongly dependent on cell type, sex and brain region.’ Also, some GABA migrate from pial during

postnatal period (Léger, et al 2020). With this said, not sure if the authors should state something is not totally proven yet for the brain region they are studying.

We agree with the reviewer that the brain region and the age have an important effect on the development of the GABA reversal potential. The reference (Murata and Colonase, 2020) shows that optogenetic/chemogenetic stimulation of GABA neurons in visual cortex have an inhibitory effect at P3 (their Figure 3). We have clarified in the text that these inhibitory effects were shown in visual cortex. We have also cited work from Babij et al. (2022) showing GABA produces inhibitory currents at P7 in the somatosensory cortex (line 50). For brevity, we prefer just to focus on what is known about the somatosensory and visual cortex in the age range we are studying here, and not focus on the hippocampus and other areas.

Methods:

1) The neonatal is a critical period for neurodevelopment. I wonder if the invasive procedures (craniotomy, electrophysiology among others) used along this study might disrupt the physiological development. Such as electrodes implantation create inflammation or disturb the vasculature development when dura is removed (P10). Did the authors evaluate the developmental milestones and weight? Could be beyond some results?

The electrophysiological results were obtained from acute experiments in a separate set of mice from the imaging studies, so none of the imaging mice would have this procedure done to them. They were euthanized immediately after the electrophysiological recordings. We have made this more clear in the Methods section.

2) The authors mention many times in the manuscript adult-like but is not clear what is the age that adult-like is referring to. Please state it in the methods. Related with this question can the authors also explain the selection of ages?

We define adult on line 98 of the results “Following a whisker stimulus, blood volume increases were observed in adult (~4 months old) mice,....”

In the Methods, line 419, we have added:

“Ages were chosen to span the period of rapid changes around eye opening (~P14) in which neural responses in the barrel cortex become more adult-like^{46,59} and to detect any more subtle changes in the following weeks.”

3) Along the methods is not clear is the study is a longitudinal or terminal experiments. Please specify. If longitudinal, or same animals used for the same experiments would be a great addition correlations within the NVC bold results and arousal status.

All electrophysiology experiments were performed as terminal experiments due to inability of subjects to be returned with litter-mates with implanted electrodes. In order to account for similar developmental trajectories and familiarity with imaging experiments, data included in Fig 1 for all ages except adult are from subjects undergoing their first imaging session.

In fMRI experiments, all mice were imaged longitudinally. However, fMRI data of some imaging points were not included due to excessive motion during imaging or animal death. In the visual stimulation experiment (n = 38), fMRI data used in the final analysis include: 5 mice imaged for all 6 time points, 6 mice imaged for 5 time points, 4 mice imaged for 4 time points, 3 mice imaged for 3 time points, 12 mice imaged for 2 time points, and 8 mice imaged one time point. In the whisker stimulation experiment (n = 26), fMRI data used in the final analysis include: 5 mice imaged for all 6 time points, 8 mice imaged for 5 time points, 2 mice imaged for 4 time

points, 4 mice imaged for 3 time points, 5 mice imaged for 2 time points, and 2 mice imaged for one time point.

This has all been added to the methods section, starting at line 423.

Results

1) In the results section will be easier to follow if the authors describe the outcomes from the younger to older, to show the normal developmental process timeline. Namely at the 'neonatal mice show frequent sleep-wake transitions'.

We have changed the text so that P10 mice are discussed first.

2) Fascinatingly P10 neurons fired indeed, while quite differently from the other age P15 or P30, first they started in almost zero Hz then after fire there is a decrease on the spike rate. I am not familiar with neuronal activity, is this a normal response or neonatal related?

The low spontaneous firing rate at young ages has been seen before (Babij et al., 2022; Che et al., 2018). We have noted this in the text (line 131).

3) Supplemental Figure 1 – the figure has P30 but the legend refers P15. 'Neural responses to whisker stimulation in a P15 mouse.' Are the results for what age?

P30, this has been corrected.

4) Regarding with the visual stimulation and visual cortex response, at P14 the animals should have the eyes open, and I was expecting a V1 BOLD response at this age. However, the results showed a small response at SC but nothing at visual. Any possible explanation for this unexpected result?

One possibility is that even though the eyes are open, the mice are asleep. Adult mice sleep with their eyes open. We have addressed this on line 374 in the Discussion.

5) If P10 spent the most of time in active sleep this should not mean the neurons would be in a more 'activated' status or not activate the NVC in the baseline?

Previous investigations (Tiriac et al. 2014, Dooley et al. 2020) have shown that motion causes increase in neural activity in young (~P5) rats, and that the firing rates during active sleep are higher than during movements. We have addressed this concern in the Discussion section, (line 375).

6) The previous literature has tested a similar approach but using anesthesia animals. What stage of sleeping or arousal this will represent comparing with your results? If P10 or P15 are in a sleeping mode, the anesthesia should not have that much impact because is causing similar vascular dilatation and the stimulation will awake the same way the animal. So, the aftermath will be the same right?

To better compare our results with unanesthetized mice with previous work in anesthetized animals, we have added the following to the Discussion section (Line 380):

“Relating our results to previous developmental studies with anesthetized animals is more complicated. In adults, sleep is associated not only with large changes in local neural activity, but also neuromodulation changes⁷⁵, and increases in noradrenergic signaling is associated

both with awakening⁷⁶ and vasoconstriction^{77,78}. There is no single anesthetized state⁷⁹, and different anesthetics have varying effects on neurons and the vasculature⁸⁰, with some anesthetics giving more sleep-like activity than others⁸¹. For example, there are similar arousal transitions during urethane as during normal NREM sleep⁸². If stimulation causes animals to transition to a more awake-like, vasoconstricted state (with higher noradrenergic tone) this would explain the similarities between our observations and those made in urethane anesthetized animals^{9,11}.”

7) *The authors mentioned that some of BOLD response might is not due developmental processes but adult habituation. Maybe I understood wrongly but all the animals are exposed to the same machine acquisition noise. If the no anesthetised P10 pup spends most of the time sleeping, I do not think the noise or habituation will have an impact on the BOLD/hemodynamics response in this study and this will be developmental. Would not be possible some habituation in pups 1 or 3 days to see if these has really an impact as the authors said.*

The reviewer is correct that all animals were exposed to the same scanner noise during data acquisition. We have removed the statement about the speculation that some of differences in BOLD response could be due to habituation as we do not have evidence to support this statement.

8) *Supplemental Figure 5 – I believe the authors instead of ‘Active asleep’ on B and C wanted to mention ‘REM sleep’ to be consistent with the remaining results. Or at least match the colors of A.*

Corrected

Discussion:

1) *There are some important results in the manuscript the authors could try discuss/explaining*

within the discussion perhaps with literature. Such as:

a) 'These results suggest that there are qualitatively different vascular dynamics in the neonatal mouse than the adult. Intriguingly if these low-frequency oscillations interact with sensory stimulation, they could greatly impact the hemodynamic response (49).'

The relevance this paper to our work is explained in the Discussion, line 391. To aid the reader, we have added 'see Discussion' at the conclusion of this sentence.

Minor comments:

1) The Figure 1 has included 'Auditory stimulus' that was never mentioned in the results. Not sure if this was a typo at least was not really tested perhaps is caveat.

See response to Reviewer 1 on this point.

2) Still in Figure 1 page 17 line 551-552 – 'Note that for the P10 age group, 4 animals lacked a clear peak and recovery in $\Delta[HbT]$ and were excluded from the FWHM and peak calculations.' Eventually could be on the methods and not on the legend. However, any explanation why this is happening?

P10 mice show little to no sensory evoked response (see Fig 1E), so in some cases the peaks are spurious and came at a much later time that we could not be sure they are sensory evoked. If we prefer to put this in the legend so that the reader will see it. It will probably be overlooked if put in the Methods.

3) Page 7 line 22 – typo in the references.

Fixed

4) Figure 1 legend: 'Imaging blood volume changes in the somatosensory cortex of neonatal mice.' Since the authors have also 4 months included at this dataset would be fairer 'Imaging blood volume developmental changes in the somatosensory cortex of mice.' Or 'Imaging blood volume changes across life-stages in the somatosensory cortex of mice.'

We have changed it to "Imaging blood volume changes in the somatosensory cortex of mice at different ages"

5) Supplemental Figure 1: What this means? 'Same as in B.'

We have clarified it as "Layout is the same as in B"

6) Page 14 line 472, typo on the units 'um'.

Fixed

Reviewer #3 (Remarks to the Author):

Overall: The authors propose disambiguating the (at least) 2 distinct hemodynamic responses likely triggered in barrel cortex on stimulating the whiskers of mouse neonates (age: P10, P15). Whisker stimulation evokes neural activity which leads to increased local blood volume, albeit an order of magnitude more slowly than in adults. At the same time, the stimulation likely wakes the neonates as they typically spend a majority of their time sleeping. This sleep to wake transition leads to a sharp decrease of local blood volume, modulating and reversing the stimulus-triggered hyperemia. The authors propose that this combination of two hemodynamic phenomena accounts for the biphasic stimulus-evoked hemodynamic response observed in neonates.

The evidence presented is compelling and the writing is clear. I have no comments other than a

few very minor copy editing points. There are a few typos (line 60: 'awakening IS causes ...'. Line 226: 'NEM' in place of 'NREM'.) etc.

Corrected

Also suggest replacing line 239/240: '50% behaviors LESS THAN 53+/- 32 s' with '... longer than ...' since the authors are drawing attention to the striking length of the active sleep bouts. This is the same phrasing as when quantifying long REM epochs in P15 (line 225/226), as vs the 'less than...' phrasing used in lines 241/242 when pointing out the striking brevity of wake epochs.

Corrected

Aniruddha Das

REVIEWERS' COMMENTS:

Reviewer #1 (Remarks to the Author):

No further comments

Reviewer #2 (Remarks to the Author):

The authors have addressed all my comments. I have no further comments and I recommend this manuscript for publication.